# Energy Drinks and Their Acute Effects on Heart Rhythm and Electrocardiographic Time Intervals in Healthy Children and Teenagers: A Randomized Trial

**DOI:** 10.3390/cells11030498

**Published:** 2022-01-31

**Authors:** Guido Mandilaras, Pengzhu Li, Robert Dalla-Pozza, Nikolaus Alexander Haas, Felix Sebastian Oberhoffer

**Affiliations:** Department of Pediatric Cardiology and Pediatric Intensive Care, Hospital of the University of Munich, Ludwig Maximilians University Munich, 81377 Munich, Germany; pengzhu.li.extern@med.uni-muenchen.de (P.L.); Robert.DallaPozza@med.uni-muenchen.de (R.D.-P.); Nikolaus.Haas@med.uni-muenchen.de (N.A.H.); felix.oberhoffer@med.uni-muenchen.de (F.S.O.)

**Keywords:** energy drinks, electrocardiography, extrasystole, heart rate, caffeine, arrhythmias

## Abstract

Beyond their effect on blood pressure, the effect of energy drinks on heart rate in children and teenagers has not been evaluated until now. Thus, this study aimed to investigate the acute cardiovascular effects of energy drinks in healthy children and teenagers. Twenty-six children and adolescents (mean age 14.49 years) received a commercially available energy drink (ED) and placebo on two consecutive days based on the maximum caffeine dosage as proposed by the European Food Safety Authority. Heart rhythm and electrocardiographic time intervals were assessed in a prospective, randomized, double-blind, placebo-controlled, crossover clinical study design. ED consumption resulted in a significantly increased number of supraventricular extrasystoles (SVES) compared to the placebo, whereas supraventricular tachycardia or malignant ventricular arrhythmias were not observed. The mean heart rate (HR) was significantly lower following consumption of EDs. In contrast, QTc intervals were not affected by EDs. Being the first of its kind, this trial demonstrates the cardiovascular and rhythmological effects of EDs in minors. Interestingly, EDs were associated with adverse effects on heart rhythm. Whether higher dosages or consumption in children with preexisting conditions may cause potentially harmful disorders was beyond the scope of this pilot study and remains to be determined in future trials. Trial Registration Number (DRKS-ID): DRKS00027580.

## 1. Introduction

Energy drinks (ED), marketed for providing mental and physical stimulation, are beverages containing stimulant compounds such as caffeine [1]. Though some of their side effects on the cardiovascular system, such as arterial hypertension, are well-known, the acceptance of such drinks remains very high, especially among teenagers. 

According to NOMISMA-ARETÉ Consortium for the European Food Safety Authority (EFSA), the highest prevalence of ED consumption was reported in adolescents (68%), followed by adults (30%) and children (18%) [2].

Corresponding to their growing popularity with minors, data suggest that emergency admissions associated with ED consumption have been increasing [3]. Excessive intake, particularly combined with party drugs and alcohol and/or in the presence of cardiovascular conditions, may more frequently lead to adverse cardiovascular events such as cardiac arrhythmia, myocardial ischemia, etc. [4]. Several clinical trials demonstrated a significant increase of arterial blood pressure after acute ED consumption in adult subjects [5,6,7,8]. A recent study from our institution indicates that the pediatric cardiovascular system potentially reacts even more severely to the acute ED intake than in adults [9]. In addition to the hemodynamic effects, ED intake may also be associated with proarrhythmic effects, i.e., a QTc prolongation in adults [6]. 

The aim of this study was to investigate the acute effects of ED consumption on heart rhythm and electrocardiographic time intervals in healthy children and teenagers by conducting a randomized, double-blind, placebo-controlled, crossover clinical trial.

## 2. Materials and Methods

### 2.1. Ethical Statement 

The study was conducted according to the guidelines of the Declaration of Helsinki and approved by the Ethics Committee of the Ludwig Maximilians University Munich (Munich, Germany) (protocol code: 20-0993, date of approval: 12 January 2021). Prior written informed consent was obtained from all study participants. In minor study participants, prior written informed consent was additionally obtained from parents or legal guardians. 

### 2.2. Study Population

Healthy children and teenagers between the ages of 10 and 18 years were prospectively recruited for this study. Study participants were examined for eligibility before enrollment through a personal interview, clinical examination, conventional echocardiography, 24 h Holter ECG, and 24 h blood pressure monitoring. 

Exclusion criteria were as follows: chronic cardiovascular conditions (e.g., congenital heart disease, arterial hypertension, severe dysrhythmia), history of sudden cardiac death within the family, allergies to beverage ingredients, regular use of medication with effects on cardiovascular function, regular use of drugs including smoking and alcohol consumption, and pregnancy. 

Inclusion criteria specified for making potential participants eligible for the study were the following: healthy children and adolescents between the ages of 10 and 18 years.

The weight classification of included participants was assessed according to body mass index (BMI, kg/m^2^) percentiles (P.) established by Kromeyer-Hauschild et al. [10]. 

General caffeine and ED consumption behavior of participants was graded in accordance with Shah et al.: rare caffeine consumer if <1 caffeine-containing drink per month, occasional caffeine consumer if 1 to 3 caffeine-containing drinks per month, frequent caffeine consumer if 1 to 6 caffeine-containing drinks per week, and daily caffeine consumer if ≥1 caffeine-containing drink per day [7].

### 2.3. Study Design

This study was a randomized, double-blind, placebo-controlled, crossover clinical trial conducted between April and October 2021 at our institution. On two consecutive days, the participants received either a commercially available caffeinated ED or a placebo drink without the conventional ingredients found in an ED (e.g., caffeine, taurine). They were randomly assigned to the two study groups (Group I: day 1: ED, day 2: placebo; Group II: day 1: placebo, day 2: ED). 

The amount of ED administered was calculated by bodyweight (3 mg of caffeine per kilogram bodyweight), representing the maximal daily dose of caffeine for minors as recommended by the European Food Safety Authority [11]. The volume of placebo drink administered was matched to the ED in milliliters. According to the product label, the ED contained caffeine (32 mg/100 mL), taurine (200 mg/100 mL), glucuronolactone (24 mg/10 mL), ginseng aroma extract (10 mg/100 mL), guarana extract (10 mg/100 mL), and vitamins. The placebo drink contained carbonated water, multi-fruit juice, as well as fruit and vegetable extracts. Both beverages had a similar sugar content (ED: 15.2 g/100 mL, placebo drink: 13.2 g/100 mL) and taste. Both beverages were administered in an identical and masked drinking bottle at room temperature on the two consecutive days. Study participants were not allowed to consume any sources of caffeine (e.g., coffee, tea, chocolate) or drugs (e.g., tobacco, alcohol) 48 h before and 24 h after participation. Overnight fasting (apart from water) was requested prior to every study day. Lastly, the participants were expected to not consume any food or liquids during each examination period. 

### 2.4. End Points 

End points were heart rate (HR, bpm), QTc interval (QTc, ms, as well as the number of supraventricular (SVES) and ventricular extrasystoles (VES). The end points were assessed for the following time periods after beverage consumption on both study days, respectively: 0–60, 60–120, 120–180, and 180–240 min. To minimize circadian rhythm changes, the beverages were administered at similar morning hours on both days [7]. Moreover, study participants were asked to stay in the supine position for the whole duration of the examination to minimize the influence of physical activity on the recorded cardiovascular parameters. 

Electrocardiographic data were recorded in patients while in the supine position by utilizing a portable 3-lead Holter ECG device (CardioMem^®^ CM 4000, getemed, Teltow, Germany). Electrodes were positioned according to conventional ECG documentation guidelines, since all included study participants displayed normal cardiac anatomy [12]. ECG data were evaluated offline by two blinded researchers. Mean HR, the heart rhythm, the presence of SVES and VES, as well as the QTc trend were analyzed. Bazett’s formula was applied for QTc interval correction (QTc, ms) [13]. Furthermore, data on the dynamic analysis of the QT duration (QT–RR relationship) were collected and analyzed. The device allows independent beat-to-beat measurements of QT and proceeding RR intervals on each lead. Recorded ECG indices were evaluated according to sex- and age-specific reference values [14].

### 2.5. Statistical Analysis 

A two-way repeated-measures ANOVA was performed to evaluate the effect of the two different beverages on mean HR, QTc, and QT–RR relationship over time. Sqrt data transformation and the Wilcoxon rank test were applied if the data were not normally distributed. For enumeration data including the number of SVES and VES, Poisson regression or negative binomial regression was implemented according to the results of the expected value and variance. The Bonferroni adjusted pairwise test was used for post hoc testing (R version 4.1.1). Data analyses were independently performed by a masked researcher. A *p*-value < 0.05 was considered as statistically significant. Being a pediatric pilot study, pediatric reference values of ED-induced changes in electrocardiographic time intervals did not exist and thus could not be considered in a power analysis.

## 3. Results

### 3.1. Patient Characteristics

In total, 26 healthy children and teenagers were included in the analysis. Participants’ characteristics are displayed in Table 1. None of the subjects had chronic health conditions or were under medicinal treatment. Twelve out of twenty-six study participants (46.15%) correctly guessed the day of ED administration, suggesting appropriate blinding quality. 

### 3.2. Heart Rhythm, Supraventricular, and Ventricular Extrasystoles 

Twenty-three study participants presented with sinus rhythm, and the other three displayed a heart rhythm alternating between sinus and junctional rhythm. None of the participants demonstrated changes or abnormalities in heart rhythm after beverage consumption. 

Negative binomial regression was applied since the mean and variance of SVES data were not equal. A significantly higher occurrence of SVES after ED consumption compared to placebo intake was demonstrated during the four-hour observation period (incidence rate ratio: 1.700 (1.058, 2.732), *p* = 0.0276) (Figure 1). 

Poisson regression displayed no significant difference in the number of VES between ED and placebo consumption (*p* > 0.05).

**Figure 1 cells-11-00498-f001:**
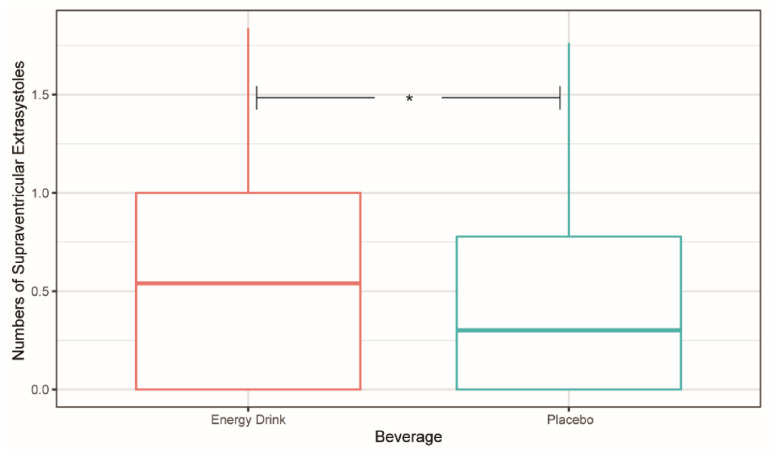
Number of supraventricular extrasystoles (SVES) within the first four hours after energy drink and placebo ingestion. The numbers of SVES were transformed by lg(*n* + 1). * *p* < 0.05.

### 3.3. Electrocardiographic Time Intervals

The Shapiro–Wilk test revealed non-normal distribution of mean HR in the placebo group for the time period 0–60 min and of the QT–RR relationship in both groups for the time periods 60–120, 120–240, and 180-240 min after consumption. Non-normally distributed data were transformed into normally distributed Sqrt-form for further analysis. After Mauchly’s spherical hypothesis test for the interaction term “beverage and time”, the variance and the covariance matrices of the dependent variables were equal (*p* > 0.05). Regarding the QT–RR relationship data, the Wilcoxon signed-rank test was applied to analyze the effect of beverage consumption on different time intervals.

### 3.4. Mean Heart Rate

The interaction between “beverage and time” had a statistically significant effect on mean HR (*p* < 0.001). Therefore, the separate effect of the variable “beverage” was analyzed at each time interval.

The mean HR was demonstrated to be lower in the ED group compared to the placebo group during the time period of 60–120 min after beverage consumption, with a mean difference of 2.71 bpm, respectively (*p* = 0.012) (Table 2). The remaining time periods did not show significant differences in mean HR between both beverages (Table 2, Figure 2).

**Table 2 cells-11-00498-t002:** The separate effect of beverage on mean heart rate (*n* = 26).

Parameters	Energy Drink (bpm)	Placebo (bpm)	*p*-Value
Time 1 h	78.77 ± 9.58	79.92 ± 9.32	0.333
Time 2 h	79.54 ± 8.85	82.65 ± 8.81	0.012 *
Time 3 h	78.31 ± 9.18	76.39 ± 7.64	0.125
Time 4 h	76.19 ± 9.14	73.85 ± 8.79	0.095

Mean ± standard deviation was used for normally distributed variables. * *p* < 0.05.

**Figure 2 cells-11-00498-f002:**
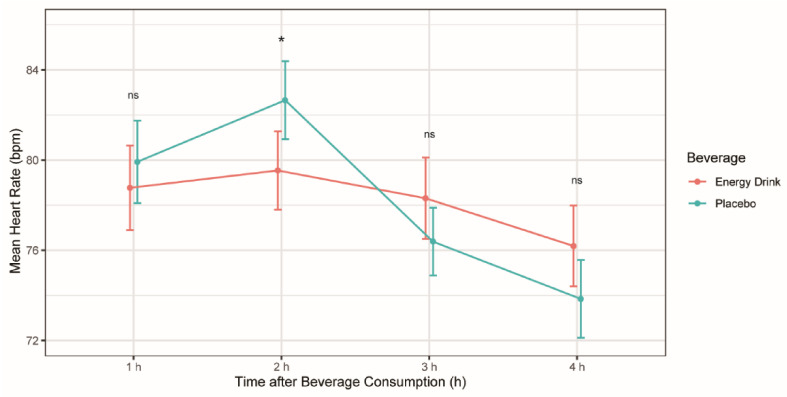
Mean heart rate (bpm) after energy drink and placebo consumption at different time periods. *p < 0,05, ns: no significant difference.

### 3.5. QTc

The interaction between “beverage and time” had no statistically significant effect on QTc lead 1 and lead 2 (*p* > 0.05). In addition, the main effect of the variable “beverage” on QTc lead 1 and 2 did not show a significant difference at any time point (*p* > 0.05) (Table 3). 

Due to the increased prevalence of artefacts in multiple study participants, lead 3 was excluded from the final data analysis. 

**Table 3 cells-11-00498-t003:** QTc lead 1 and QTc lead 2 (*n* = 26).

Parameters	QTc Lead 1		QTc Lead 2	
	Energy Drink (ms)	Placebo (ms)	Energy Drink (ms)	Placebo (ms)
Time 1 h	429 ± 23.6	428 ± 27.0	427 ± 25.0	429 ± 28.3
Time 2 h	428 ± 22.6	429 ± 26.0	429 ± 20.9	432 ± 26.9
Time 3 h	427 ± 22.5	425 ± 22.5	426 ± 22.5	430 ± 24.1
Time 4 h	432 ± 23.6	429 ± 24.3	434 ± 24.3	434 ± 25.6

Mean ± standard deviation was used for normally distributed variables.

### 3.6. QT–RR Relationship

For the time period 60–120 min after beverage consumption, the Wilcoxon signed-rank test revealed a significantly lower QT–RR relationship in the ED group compared to the placebo group, with a median difference of 0.062 (*p* < 0.01). No significant difference in the QT–RR relationship was assessed for the remaining time periods between ED and placebo groups (Table 4, Figure 3).

**Table 4 cells-11-00498-t004:** The effect of beverage on the QT–RR relationship (*n* = 26).

Parameters	Energy Drink	Placebo	Difference	*p*-Value
Time 1 h	0.683	0.646	0.047	0.069
Time 2 h	0.655	0.700	−0.062	0.007 **
Time 3 h	0.695	0.685	−0.017	0.316
Time 4 h	0.602	0.561	0.049	0.443

Median was used for non-normally distributed variables. ** *p* < 0.01.

**Figure 3 cells-11-00498-f003:**
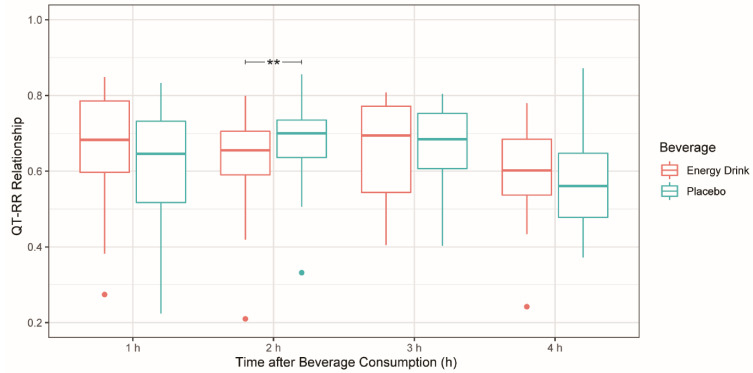
QT–RR relationship after energy drink and placebo consumption at different time periods. ** *p* < 0.01.

## 4. Discussion

Medical associations, including the American Academy of Pediatrics and the American Medical Association, advise against ED consumption in minors due to the potential health risks associated with these beverages [15,16,17]. Although children and teenagers represent one of the main ED consumer groups, the acute effects of ED consumption on the pediatric cardiovascular system have not yet been evaluated. 

The dosage of the administered EDs was based on recommendations provided by the EFSA, suggesting a maximum intake of up to 3 mg of caffeine per kilogram of bodyweight as appropriate for healthy children and adolescents [11]. Even though there have been several case reports connecting caffeine consumption with cardiac arrhythmias, large-scale population studies have not been able to validate these associations [18].

To the best of our knowledge, this is the first study investigating the acute effects of ED consumption on heart rhythm and electrocardiographic time intervals in healthy children and teenagers. A randomized, double-blind, placebo-controlled, crossover study design was used to ensure data validity. In total, 26 healthy children and teenagers with a mean age of 14.49 years were included in this study. 

### 4.1. Energy Drinks and Their Acute Effects on Heart Rhythm and Electrocardiographic Time Intervals 

The arrhythmogenic potential of caffeine is controversially discussed in the literature [19]. In a review by Goldfarb et al., 17 cases of acute cardiovascular events related to ED consumption in individuals between 13 and 58 years—15 thereof under 30 years—were identified. The cardiovascular events described varied from atrial and ventricular arrhythmias to QT-prolongation, ST-segment elevations, and cardiac arrest. In five of these cases, acute heavy ED consumption was described, four reported co-ingestions with alcohol or other party drugs, and two were found to have a channelopathy [20]. These adverse effects may be associated with the high caffeine content in EDs. Nevertheless, the relevance of these data may be limited by the retrospective data collection and the inhomogeneous study cohorts. Contrary to the results of our prospective study design, the presence of illegal substances, prior chronic caffeine consumption before the ED intake, as well as the comorbidity with other cardiovascular risk factors cannot be excluded in those data and may have potentiated the arrhythmogenic effect of the EDs. An invasive prospective placebo-controlled, randomized, electrophysiological study performed by Lemery et al. found that caffeine, at moderate intake (5 mg/kg), was associated with a significant increase in systolic and diastolic blood pressure but had no significant effect on cardiac conduction and refractoriness. Furthermore, caffeine intake at a moderate level showed no effect on supraventricular tachycardia induction or other forms of tachycardias [8]. 

In the present trial, the consumption of an ED containing the amount of caffeine suggested by the EFSA led to a significant increase in the total number of SVES compared to the placebo in minors without any prior rhythm pathologies or any risk factors for dysrhythmias. When premature, they may be blocked or conducted aberrantly, causing fatigue, dyspnea, or syncope. Furthermore, SVES may lead to supraventricular tachycardia with atrial fibrillation and a consecutive stroke may be induced [21,22,23]. Whether the elevated number of ectopic beats is a result of caffeine consumption or of an increased vagal tonus should be the subject of further research.

In contrast to studies on EDs performed in adults, no statistically significant QTc alteration was found in the present pediatric pilot study [7,24,25]. One could speculate that this observation might be due to the lower caffeine dosage administered in our study as compared to adults.

Our study also revealed a significant decrease in the mean HR in the time period 60–120 min following ED consumption, as opposed to most studies in adults which demonstrated an increase in HR after the consumption of caffeinated EDs [24,25].

This study also addressed a possible arrhythmogenic risk (QT prolongation) of acute ED consumption by analyzing the QT–RR relationship. A significant decrease in the time period 60–120 min after ED intake was observed. 

Previous data suggest that the steepness of the QT–RR relationship depends on HR [26]. A decrease in HR leads to higher RR intervals and thus to a flattened QT–RR relationship [27]. Hence, the significantly decreased QT–RR relationship shown in this study may be explained by the significantly lower HR observed.

Whether the decrease of HR and of the QT–RR relationship has pathophysiological effects on the cardiovascular system remains to be determined. Preliminary results from our institution demonstrated that the arterial stiffness of the right common carotid artery increased significantly two hours after ED consumption compared to the placebo intake in 19 of the participants included in the present study [9]. Taking these data into consideration, the elevated arterial stiffness caused by sympathetic stimulation due to caffeine may lead to an increased blood pressure. This mechanism may subsequently initiate a reflexive decrease in HR and the QT–RR relationship through baroreceptor and parasympathetic stimulation. The dynamic state of the QT–RR relationship could be influenced by autonomic change. A normal dynamic QT–RR relationship connects autonomic reflex responses such as tachycardia and bradycardia with QT hysteresis (the lag between QT adaptations for a given RR interval change). QT alterations may represent a caffeine-induced altered repolarization associated with yet undefined arrhythmogenic risks [28]. Therefore, further research is required evaluating vagal activation and its potential impact on electrocardiographic time intervals after the intake of caffeinated EDs. 

### 4.2. Energy Drinks: Potential Health Threat for Children and Teenagers?

The European Cardiac Arrhythmia Society suggests that children under 14 years of age and children with underlying cardiac conditions should refrain from consuming caffeinated products, and suggests for children over 14 years a maximal daily caffeine dosage of 2.5 mg per kg of bodyweight [17]. This study demonstrated that the intake of caffeinated EDs with a dosage of 3 mg of caffeine per kilogram of bodyweight is associated with acute increase of SVES in a population of healthy children and adolescents. Therefore, the upper caffeine level suggested by the EFSA should be critically discussed and further research should be performed.

### 4.3. Limitations

The significance of the present study could be limited due to the relatively low number of participants (*n* = 26). Although the ED and placebo used in this study were similar in taste and were administered in an identical and masked drinking bottle, some study participants may have identified the given beverage by taste, smell, or physical response. We regard the blinding quality to be appropriate, as only 46.15% of study participants correctly guessed the day of ED administration. One specific ED product was used for this study. The pediatric cardiovascular system might respond differentially to greater ED amounts, different ED products, or to the combination of EDs with alcohol or party drugs. Additionally, solely healthy children and teenagers were included in this study. Minors with cardiovascular conditions (e.g., long-QT syndrome) might respond in a different pathophysiological manner after ED consumption. Furthermore, this study only assessed the acute cardiovascular effects of ED consumption. The impact of chronic ED consumption on the pediatric cardiovascular system has not been studied yet and requires further research. 

## 5. Conclusions

Acute ED consumption was associated with a significantly increased number of SVES in healthy children and teenagers. Furthermore, a significant decrease in HR possibly caused by an acute spike of both the systolic and diastolic blood pressure was observed. Although no significant QTc alterations were detected after ED intake compared to placebo, there was a significant decrease in the QT–RR relationship, indicating a reflective autonomic response. The current findings suggest that minors suffering from heart rhythm conditions might develop malignant dysrhythmias after consuming these beverages. Further investigations of the activation of the sympathetic and parasympathetic nervous system by EDs along with the possible clinical effects this pathway may have on the cardiovascular system should be conducted.

## Figures and Tables

**Table 1 cells-11-00498-t001:** Study participants’ characteristics (*n* = 26).

Characteristics	Total
Age (years) mean (SD)	14.49 ± 2.44
Sex, *n* (%)	
Male	13 (50)
Female	13 (50)
Weight Classification, *n* (%)	
Normal weight	22 (84.62)
Overweight	4 (15.38)
Obese	0 (0)
Caffeine Consumption Behavior, *n* (%) ^a^	
Rarely	16 (61.54)
Occasionally	3 (11.54)
Frequently	5 (19.23)
Daily	2 (7.69)
Energy Drink Consumption Behavior, *n* (%) ^b^	
Never	11 (42.31)
Rarely	11 (42.31)
Occasionally	1 (3.84)
Frequently	3 (11.54)
Daily	0 (0)

^a^ Rare caffeine consumer if <1 caffeine-containing drink per month, occasional caffeine consumer if 1 to 3 caffeine-containing drinks per month, frequent caffeine consumer if 1 to 6 caffeine-containing drinks per week, and daily caffeine consumer if ≥1 caffeine-containing drink per day [7]. ^b^ Rare energy drink (ED) consumer if <1 ED per month, occasional ED consumer if 1 to 3 EDs per month, frequent ED consumer if 1 to 6 EDs per week, and daily ED consumer if ≥1 ED per day.

## Data Availability

The data presented in this study are available upon request from the corresponding author.

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
