# Peer review of "Energy Drinks and Their Acute Effects on Heart Rhythm and Electrocardiographic Time Intervals in Healthy Children and Teenagers: A Randomized Trial"

_cells, 2022, doi:10.3390/cells11030498_

Round 1
Reviewer 1 Report
The topic of this article is of high interest, but minor changes are necessary in order to be published.
- My suggestion is to specify whether the results of ambulatory blood pressure monitoring presented in the discussion section belong to the present group of patients or to other group of patients (line 223-228). If these results belong to this group, I suggest to present more details about the results.
- The reference number 9 has not yet been published and I suggest to be retracted.
Reviewer 2 Report
the authors presented a study on the acute effect of energetic drinks on childern. The matter is worthy of interest due the increased consuption of these beverages all around the world.
The study is well designed and limitations are crearly indicated. The manuscript is well written and fully intelligible.
Results are quite limited due the low number of partecipants and the short term follow-up.
However, there are some issues the authors should address
1- they mentioned a previous study conducted by the same group. (reference 9). However, this study is not yet published due they stated it is still under review. Furthermore, The study is not published as a preprint or something similar. In this form this study must not be mentioned. Thus the reference must be delated and the relative part of main text must be changed.
2- In discussion a review of cases is reported (reference 20: Goldfarb, M.; Tellier, C.; Thanassoulis, G. Review of Published Cases of Adverse Cardiovascular Events After Ingestion of Energy Drinks. The American Journal of Cardiology 2014, 113 (1), 168–172. https://doi.org/10.1016/j.amjcard.2013.08.058. This reference is not directly comparable to the study the authors performed because it is a review and include also patients who were assuming drugs. I suggest to be more strong in indicating the differences between their results to this reference
3- The authors presented the data about QT-RR. They did not explained what QT-RR is, how they evaluated, why they are doing this evaluation or whether it important. I guess they evaluated QT/RR as a marker of rhythm variability but more information are needed
4- there is a difference according to sex distribution?
5- QTc values should be provided. I understand it is not significantly different between the two groups but values should be indicated at least in main text.
6- discussion is quite poor compared to introduction. This may be due also to the few results presented. I suggest to increase discussion in order to increase also the value of the manuscript. In example, just few words were spent on the main finding (the increased number of supraventricular extrasystoles - SVES - the authors had found).
7- Similar to previous comment, the authors must considered to increase discussion on QTc. In fact, QTc findings in this study is apparently different to the cited literature
Round 2
Reviewer 2 Report
The authors addressed all the queries. Good job